# Facile access to benzofuran derivatives through radical reactions with heteroatom-centered super-electron-donors

Shichun Jiang[1,4], Wei Wang[1,4], Chengli Mou[2,4], Juan Zou[2], Zhichao Jin[1] ✉, Gefei Hao[1] ✉ & Yonggui Robin Chi [1,3] ✉

The development of suitable electron donors is critical to single-electron-transfer (SET) processes. The use of heteroatom-centered anions as super-electron-donors (SEDs) for direct SET reactions has rarely been studied. Here we show that heteroatom anions can be applied as SEDs to initiate radical reactions for facile synthesis of 3-substituted benzofurans. Phosphines, thiols and anilines bearing different substitution patterns work well in this intermolecular radical coupling reaction and the 3-functionalized benzofuran products bearing heteroatomic functionalities are given in moderate to excellent yields. The reaction mechanism is elucidated via control experiments and computational methods. The afforded products show promising applications in both organic synthesis and pesticide development.

Benzofuran is among the top 100 ring structures that exist in human clinical drugs[1]. Especially, the 3-functionalized benzofuran has been frequently found as the core structure in various natural and non-natural molecules with proven biological activities (Fig. 1a)[2–5]. For instance, amiodarone is a synthesized human drug that has been used in clinical treatment for ventricular and supraventricular arryhythmias[6–8]. 3-Substituted benzofurans are also the key fragments in the fermentation abstracts of the fungi *Talaromyces amestolkiae YX1* with inhibitory effects on the α-glucosidase in living organisms[9]. Therefore, the development of novel and efficient synthetic methods for quick and selective access to 3-substituted benzofurans bearing diverse functionalities is attractive.

To date, a diversity of synthetic approaches have been developed for efficient construction of 3-functionalized benzofuran molecules (Fig. 1b)[10–16]. Early in the 1950s, Whitaker and co-workers have developed an acid-promoted dehydrative cyclization process of electron-rich o-hydroxybenzylketones for access to a series of 3-substituted benzofuran molecules bearing a 5-phenylsulfonylamino group (Fig. 1b, Eq. 1)[17]. Nicolaou and co-workers disclosed a cascade cyclofragmentation-release strategy in 2000 for effective construction of 3-arylbenzofurans through the introduction of an epoxide moiety

onto the benzene ring of the phenyl ether substrate (Fig. 1b, Eq. 2)[18]. She and co-workers succeeded in the application of the easily accessible 2-iodophenol allyl ethers as the reaction starting materials for the preparation of 3-alkylbenzofurans through Pd-catalyzed intramolecular Heck reactions (Fig. 1b, Eq. 3)[19]. Recently, Yang, Zhang, Walsh, and co-workers developed an intermolecular radical coupling cascade reaction between the 2-iodophenyl allenyl ether and the ketimine substrates for the synthesis of a set of benzofurylethylamines in good to excellent yields (Fig. 1b, Eq. 4)[20]. Notably, the 2-azaallyl anion generated from the deprotonation of the ketimine substrate behaved as a super electron donor (SED) to initiate the radical process in this transformation.

Single electron transfer (SET) reactions have been developed for decades and represent one class of the most efficient methods for the construction of challenging structures[21–26]. The development of suitable electron donors is critical to SET processes and has attracted considerable attention[27–39]. The application of metals in low oxidation states[40,41], electrochemical reduction at a cathode[42–46], and photochemically assisted electron transfer strategies[47–50] have been extensively explored for the generation of radical species to initiate SET transformations. In addition, a diversity of electro-neutral organic

[1]National Key Laboratory of Green Pesticide, Key Laboratory of Green Pesticide and Agricultural Bioengineering, Ministry of Education, Guizhou University, Guiyang 550025, China. [2]Guizhou University of Traditional Chinese Medicine, Guiyang 550025, China. [3]School of Chemistry, Chemical Engineering, and Biotechnology, Nanyang Technological University, Singapore 637371, Singapore. [4]These authors contributed equally: Shichun Jiang, Wei Wang, Chengli Mou. ✉e-mail: zcjin@gzu.edu.cn; gefei_hao@foxmail.com; robinchi@ntu.edu.sg

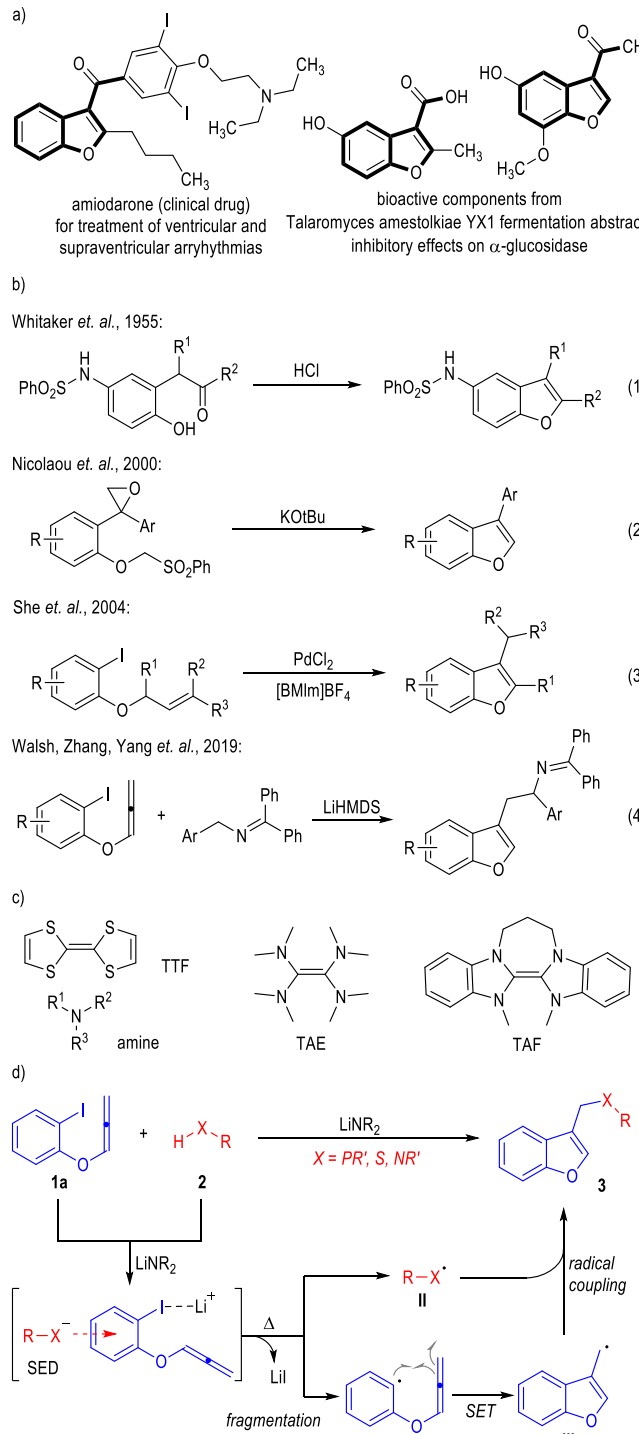

**Fig. 1 | Bioactivities, syntheses of 3-substituted benzofurans and SEDs for radical reactions. a** Commercial drugs containing 3-substituted benzofuran structures. **b** Typical methods for access to 3-substituted benzofurans. **c** Representative organic small molecular SEDs. **d** Heteroatom anions as SEDs for 3-heteroalkylbenzofuran synthesis (This Work).

molecules can also be used at their ground states as electron donors for SET reactions with conventionally inert electron acceptors (Fig. 1c)[51]. For example, early in the 1970s, tetrathiafulvalene (TTF)[52–55], tetraaminoethylene (TAE)[56–58] and their derivatives have been recognized as strong electron donors for SET transformations. Amines are also important reducing agents that can transfer a single electron to oxidative reactants to initiate radical reactions in both enantioselective

and non-asymmetric fashion[59–61]. Since 2005, Murphy and co-workers have used tetraazafulvalene (TAF) and its derivatives as SEDs for a diversity of SET reactions[62–85]. To the best of our knowledge, the use of heteroatom-centered anions as SEDs for direct SET reactions with conventionally inert benzene derivatives have not been disclosed. It is also worth mentioning that the development of general and efficient methods for access to benzofuran molecules bearing 3-phospha-/thio-/aza-methyl groups has not been reported.

Herein, we report the application of heteroatom anions as SEDs for the preparation of 3-substituted benzofuran molecules bearing various heteroatomic functionalities (Fig. 1d). The 2-iodophenyl allenyl ether **1a** is used as the reaction substrate to react with the heteroatomic compound bearing a free hydro-heteroatom (H−X) bond through radical coupling processes. The benzofuran product bearing a 3-phospha-, thio-, or aza-methyl group is afforded in generally moderate to excellent yields through this radical transformation. Mechanistically, the X-H bond of the heteroatom molecules is deprotonated by the strong base[86] and gives the heteroatom anions, which can react as SEDs to transfer one electron to the Lewis acidic lithium-activated aryliodide substrate **1a**. Then two neutral radicals **I** and **II** are generated with the elimination of one equivalent of LiI from the substrates. The phenyl radical **I** can cyclize with the allenyl group through a radical addition process and generate the benzofurylmethyl radical **III**. The heteroatom-centered radical **II** can be coupled with the radical **III** to afford the final product **3** through a radical coupling process. Noteworthily, many of the benzofuran products bearing various 3-heteroatomic functionalities we obtained from this approach exhibit good anti-bacterial activities against several plant pathogens and are promising in the development of novel pesticides for plant protection.

## Results and discussion
### Reaction development
The 2-iodophenyl allenyl ether **1a** and the diphenylphosphine (HPPh₂) were selected as the model substrates to search for a suitable reaction condition for this SED-initiated intermolecular radical coupling process (Table 1). Since the benzofurymethyldiphenylphosphine product **2a** is not stable at the ambient atmosphere, the reaction was quenched with H₂O₂ to transform the phosphine product **2a** into the corresponding phosphine oxide **3a** for isolation as the final product. Various strong bases such as *t*BuOK, KHMDS, and NaHMDS were used to deprotonate the HPPh₂, but the desired final product **3a** could only be isolated in poor yields (Table 1, Entries 1 to 3). To our great delight, the lithium amides such as LiHMDS and LDA showed extraordinary good activities in this transformation and gave the target product **3a** in satisfactory isolated yields (Entries 4 and 5). This is probably due to the increased acidity of the positively charged Li⁺ that existed in the reaction system for a better activation of the phenyliodide substrate **1a**. Other organic or inorganic bases with weaker basicities than the lithium amides were not effective for this radical coupling process (e.g., Entry 6). The reaction can be carried out in a variety of polar organic solvents, although the yields were dropped in these cases (e.g., Entries 7 to 11). Halogenated solvents such as dichloromethane were not suitable for this transformation (e.g., Entry 12). Finally, the yield of the final product **3a** could be promoted to 80% when carrying out the radical coupling reaction under 50 °C with LDA used as a base in DME (Entry 13). Further increasing the reaction temperature cannot give any improvement to the product yield (e.g., Entry 14).

### Reaction scope
With optimal reaction conditions at hand, we then examined the scope of this radical coupling reaction with aryl allenyl ethers **1** bearing different substituents (Fig. 2a). Both electron-donating and electron-withdrawing groups can be introduced to the benzene ring of the 2-iodophenyl allenyl ether substrate, with the target benzofuran

## Table 1 | Optimization of reaction conditions[a]

| Entry | Base | Solvent | T (°C) | Yield (%)[b] |
|---|---|---|---|---|
| 1 | tBuOK | DME | 30 | 16 |
| 2 | KHMDS | DME | 30 | <5 |
| 3 | NaHMDS | DME | 30 | 17 |
| 4 | LiHMDS | DME | 30 | 71 |
| 5 | LDA | DME | 30 | 76 |
| 6 | Cs$_2$CO$_3$/NaOH/ DBU/DIEA, etc | DME | 30 | 0 |
| 7 | LDA | THF | 30 | 65 |
| 8 | LDA | CH$_3$CN | 30 | 22 |
| 9 | LDA | MTBE | 30 | 20 |
| 10 | LDA | DMF | 30 | 27 |
| 11 | LDA | DMSO | 30 | 60 |
| 12 | LDA | CH$_2$Cl$_2$ | 30 | 0 |
| 13 | LDA | DME | 50 | 80 |
| 14 | LDA | DME | 60 | 74 |

[a]Unless otherwise specified, the reactions were carried out using **1a** (0.10 mmol), HPPh$_2$ (0.20 mmol), base (0.20 mmol), solvent (2.0 mL) for 8 h under N$_2$. H$_2$O$_2$ (0.25 mmol) was added into reaction system at 0 °C, then the system was stirred at 30 °C for 1 h. [b]Isolated yield of **3a**. DME = 1,2-Dimethoxyethane. HMDS hexamethydisilylamine, THF tetrahydrofuran, MTBE methyl tert-butyl ether.

products **3** afforded in moderate to good yields (Figs. 2a and 3b–j). Substituents can also be installed on the allenyl moiety of the ether substrate (**3k** to **3m**). For instance, the 1-methyl-1-allenyl alcohol ether substrate could give the target product **3k** in an excellent yield. Switching the 1-methyl group on the allenyl alcohol moiety to longer alkyl groups resulted in drops in the product yields (e.g., **3l** to **3m**). LiHMDS is sometimes needed instead of LDA as the base for this intermolecular radical coupling reaction in order to achieve constant and satisfactory product yields. It is worth noting that switching the aryl iodide substrate **1** into aryl bromides or chlorides resulted in only trace formation of the desired radical coupling products.

After examinations of the substitution patterns on the aryl allenyl ether substrates, we were interested in the search for suitable heteroatom-centered radical precursors for this SED-initiated radical coupling process. We were delighted to find that the thiols **4** also worked well in the intermolecular radical coupling process with the 2-iodophenyl allenyl ether **1a** (Fig. 2b). With a slight adjustment on the reaction condition, we successfully applied a broad scope of thiols bearing various substitution patterns in this transformation, with the corresponding 3-thiomethylbenzofuran products **5** afforded in moderate to excellent yields. For example, the thiophenols bearing electron-donating substituents on the 2- or 3-positions gave the benzofuran products **5** in excellent yields (Fig. 2b, **5a**, **5b**, **5f**, **5g**), and the ones bearing electron-withdrawing groups on the same positions only gave the target products in moderate yields (**5c** to **5e**, **5h**). Substituents on the 4-positions of the thiophenol substrates led to drops on the reaction yields (**5i** to **5k**). Naphthyl and heteroaryl mercaptan substrates generally gave the desired benzofuran products in moderate to good yields (**5l** to **5o**). Benzylthiols can also work well in this radical coupling process, although the yields of the products were only moderate (**5p** to **5q**). To our great delight, aliphatic mercaptans bearing either chained or cyclic structures worked smoothly in this radical coupling process and gave the clean

products of 3-alkylthiomethylbenzofurans in good to excellent isolated yields (**5r** to **5y**). Noteworthily, a disulfide intermediate could be observed at the beginning of this radical coupling reaction, which might be resulted from the homo-coupling of the generated sulfur radical intermediates in this transformation (for details, see supporting information).

Additionally, primary and secondary amines **6** are also promising reactants as the nitrogen-centered radical precursors for this SED-initiated radical coupling reaction (Fig. 2c). Anilines bearing both electron-donating and electron-withdrawing substituents on the 4- and 3-positions of the benzene rings could react with the 2-iodophenyl allenyl ether **1a** under similar reaction conditions to give the 3-azamethylbenzofuran products **7** in moderate yields (Fig. 2c, **7a** to **7e**). Substitutions on the 2-position of the aniline substrate **6** led to significant drops on the product yields (e.g., **7f, 7g**), which might have resulted from the steric hindrance on the nitrogen radicals caused by the 2-substituents. It is pleasing to find that an N-methyl group is also well tolerated on the aniline substrate **6**, with the target benzofuran product **7h** afforded in a higher isolated yield. Moreover, the secondary amine substrates **6** bearing various substituents and substitution patterns on the N-phenyl groups worked smoothly in this radical coupling process and led to the formation of the N,N-disubstituted aminomethylbenzofuran products in moderate yields (**7i** to **7l**). However, switching the N-methyl group of the N-methyl aniline substrate into an N-phenyl group (e.g., use diphenylamine as the substrate) could block the radical coupling process, with the corresponding benzofuran product **7m** afforded in only a poor yield under the currently optimized reaction condition.

It is worth noting that the secondary amine products **7a** to **7f** still possess a free NH group and can undergo a second nitrogen radical coupling reaction with substrate **1a**. For example, the secondary amine product **7a** can react with **1a** under the same reaction condition to give the tertiary amine product **7a'** in 20% yield.

Alcohols and phenols are also examined as the potential radical precursors for this transformation. However, none of them gave the desired benzofuran products. We assume that the oxygen radicals are not as stable as the phosphine-, sulfur-, or nitrogen-centered radicals and cannot be generated or live long enough to react with the aryl-allene radical intermediate **I** in the reaction system.

Gratifyingly, the current intermolecular radical coupling reactions can be carried out at gram scales without erosion on the product yields (Fig. 2d). The phosphine oxide **3a** can be obtained in an excellent 90% yield from 1 gram of the 2-iodophenyl allenyl ether **1a**. Alternatively, BH$_3$•THF could be used instead of the H$_2$O$_2$ in the second step of the reaction between **1a** and HPPh$_2$, with the stabilized phosphine borane complex **8** isolated as the final product in 81% yield. Moreover, the thioether product **5a** can also be afforded in a higher yield at gram scales through this protocol. Through gram-scale reactions between the secondary amine substrate N-methyl-N-phenylamine (PhNHCH$_3$) and the 2-iodophenyl allenyl ether **1a**, the 3-azamethylbenzofuran product **7g** can be afforded in a similar 58% yield.

## Mechanistic study

To probe the reaction mechanism, the detailed energy profile of the reaction processes between the phenyliodide **1a** and the HPPh$_2$ was elucidated through DFT calculations (Fig. 3). Gaussian 16[87] with DFT method M06-2×[88,89] has been employed to explore the energy variations of different transition states and intermediates throughout the radical reaction process. The normal level 6-31 G(d, p)/Lanl2dz was used for geometry optimization (6-31 G(d, p) is for C, H, O, Li, and P; Lanl2dz is for I) and the high level 6-311 + +G(2d,2p)/Lanl08 was calculated energies of a single point (6−311++G(2d,2p) is for C, H, O, Li, and P; Lanl08 is for I)[90–97]. Moreover, all calculations were under the condition of dispersion correction (D3)[98]. The HPPh$_2$ can be easily deprotonated by LDA to give the phosphine anion intermediate Ph$_2$P⁻, which

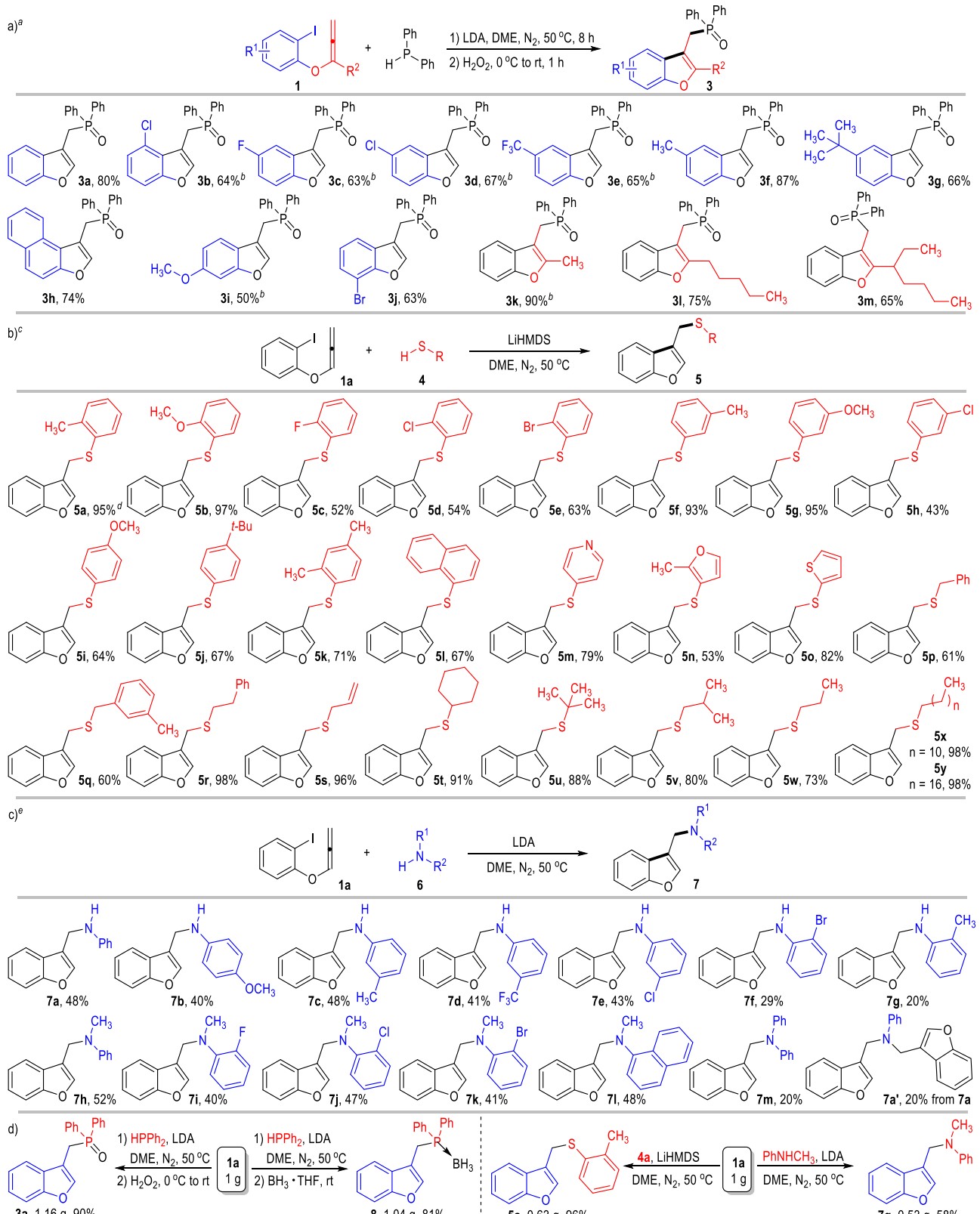

**Fig. 2 | Substrate scope and gram-scale reactions. a** Scope of the 2-Iodophenyl Allenyl Ethers **1**. **b** Application of the Thiols **4** in the radical coupling reaction. **c** Application of the anilines **6** in the radical coupling reaction. **d** Gram-scale radical reactions. [a]Reaction conditions as stated in Table 1, entry 11. Yields are isolated yields after purification by column chromatography. [b]LiHMDS (0.20 mmol) was used instead of the LDA under otherwise identical conditions to Table 1, entry 11. [c]Unless otherwise specified, the reactions were carried out using **1a** (0.15 mmol), **4**

(0.10 mmol), LiHMDS (0.20 mmol), DME (2.0 mL) at 50 °C for 12 h under N₂. Yields are isolated yields after purification by column chromatography. [d]The yield of **5a** did not change when carrying out the reaction in dark condition. [e]Unless otherwise specified, the reactions were carried out using **1a** (0.10 mmol), **6** (0.10 mmol), LDA (0.20 mmol), DME (2.0 mL) at 50 °C for 24 h under N₂. Yields are isolated yields after purification by column chromatography.

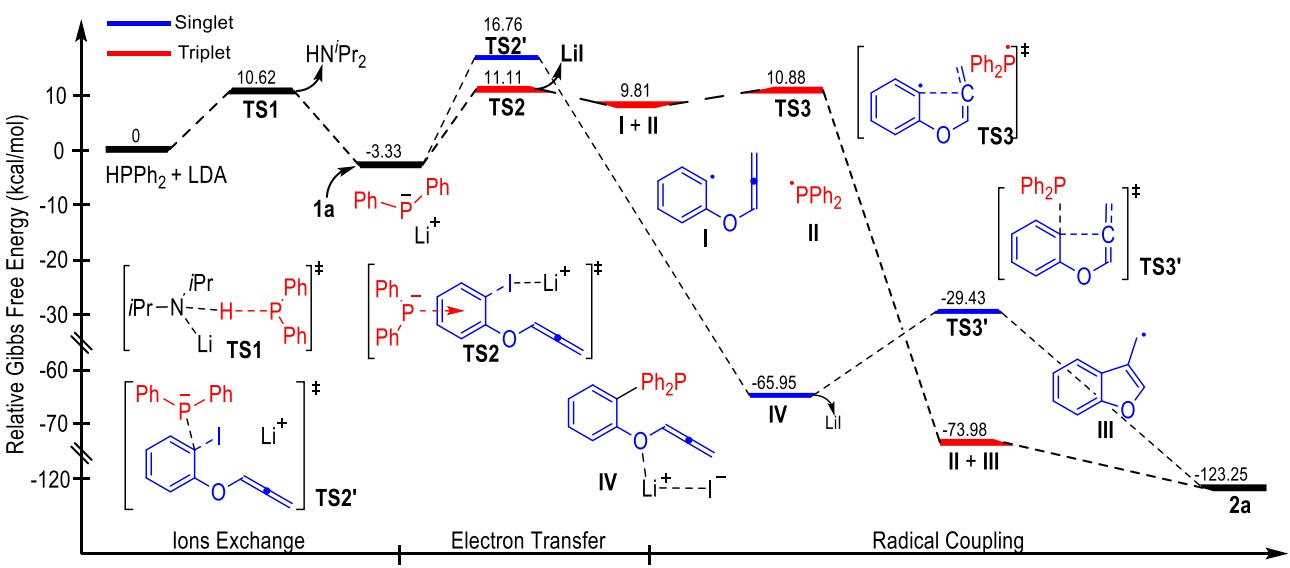

**Fig. 3 | Energy profile for the radical coupling reaction between 1a and HPPh₂.** Red lines: triplet state pathway. Blue lines: singlet state pathway.

can transfer one electron to the phenyliodide substrate **1a** via the formation of a triplet state transition state **TS2** ($\Delta G = 14.44$ kcal/mol, red lines). The afforded radical intermediate **I** is not stable and can quickly give the thermodynamically favored radical intermediate **III** via the transition state **TS3** ($\Delta G = 1.07$ kcal/mol). A spontaneous radical coupling reaction between the radical intermediates **II** and **III** gives the formation of the benzofuran product **2a**.

Alternatively, an ion-pair transfer process via a singlet state pathway is also calculated (Fig. 3, blue lines). The energy barrier for the formation of the ion pair intermediate **TS2'** is 5.65 kcal/mol higher than the triplet state transition state **TS2**. Therefore, the formation of the product **2a** through the singlet state pathway is less likely.

Although our trials in the preparation of the substrates for radical clock experiments failed, the generation of the radical species in the reaction system could be supported by the electron paramagnetic resonance (EPR) experiments (Fig. 4a). The EPR spectrum of the mixture of **1a**, HPPh₂ and LDA in DME at 25 °C exhibited a signal at approximately $g = 2.0023$, which is similar to the signal of the $g$ factor of phenyl radical.

Since both of the amino bases[99–106] and the heteroatom anions[107–114] are potential SEDs to provide single electrons to the radical acceptor **1a**, control experiments were carried out to clarify the electron donors in the radical reactions between **1a** and various heteroatom compounds (Fig. 4b). The lithium diphenylphosphanide **9** was prepared, isolated and used as a pure solid to react with the phenyliodide **1a** without the addition of any amino-containing bases (Fig. 4b, Eq. 5). The target product **3a** could be afforded in 74% yield, which clearly indicated that the phosphine anion had reacted as the electron donor in this transformation. Similarly, the lithium sulfide **10** was prepared and subjected to this radical coupling process (Fig. 4b, Eq. 6). However, the target thioether product **5a** could not form in this case, which indicated that the thio-anion might not be the real radical donor for this reaction. The secondary amine product **7a** could be smoothly afforded from the reaction between **1a** and the lithium amide **11** (Fig. 4b, Eq. 7), which supported that the amide anions, either generated from the deprotonated amine substrates or from the amide-containing basic additives, could be directly adopted as the electron donors for the current radical coupling reactions.

Additionally, two cross-radical coupling reactions between the substrate **1a** and the mercaptan **4a** were examined in the presence of the lithium diphenylphosphanide **9** and the lithium amide **11** respectively (Fig. 4c). With the lithium diphenylphosphanide **9** used as the

basic additive, both the thioether product **5a** and the phosphine product **3a** could be afforded in moderate yields (Fig. 4c, Eq. 8). In contrast, with the lithium amide **11** used as the basic additive, only trace amount of the thioether product **5a** was formed without any secondary amine product **7a** observed (Fig. 4c, Eq. 9).

The above results of the control experiments indicated that both the diphenylphosphanide and the amide anions could react as the SEDs to directly transfer one electron to the aryliodide **1a** to initiate the radical coupling reactions. The sulfide anions might be oxidized by either the amino radicals or phosphine radicals that are generated in the reaction mixture to afford the thio radicals for further radical coupling reactions.

Cyclic voltammetry (CV) studies were also carried out to test the oxidation potentials of the heteroatom-centered anions (Fig. 5). The oxidation potentials of the lithium diphenylphosphanide **9** and the lithium amides (**11**, LDA and LiHMDS) are comparable to the oxidation potentials of the classic SED molecules such as **12**, TAE and TAF (Table 2, Entries 1–4 vs. Entries 7–9). Note that, the SED compound of TAF is reported to be efficient in the reduction of aryl iodides via SET processes[85], while the TAE has been directly used in the SET reduction of alkyliodides[115–118]. The oxidation potential of lithium sulfide **10** is too high to react as an SED substrate (Entry 5). Although the oxidation potentials of the lithium salts were less negative than the reduction potential of the substrate **1a**, the irreversibility of the elimination of the iodide anion from **1a** contributed to the driving force for the SET reaction from the P- and N-centered anions to the substrate **1a**.

Based on the results of the mechanistic studies stated above, we proposed a reasonable reaction mechanism for the radical generation/coupling processes as depicted in Fig. 6. The diphenylphosphine could be deprotonated by the lithium amide to give the lithium diphenylphosphanide **9**, which could form a non-covalent ionic complex **A** for electron transfer reactions. SET reaction and elimination of the LiI of the ionic complex A gives the neutral radical intermediates **I** and •PPh₂, which could be coupled to afford the benzofuran product **3a**.

Similarly, the amide anions that existed in the reaction mixture could also react as the SEDs to transfer one electron to the phenyliodide substrate **1a** to generate the amine radical •NR¹R² and the phenyl radical **I**. The radical coupling reaction between •NR¹R² and **I** gives the amino-containing benzofuran product **7**.

In case the mercaptan substrate **4** is presented, a hydrogen-atom-transfer (HAT) reaction could happen between the radical •NR¹R² and the mercaptan substrate **4** to give the more stable thio radical RS• and

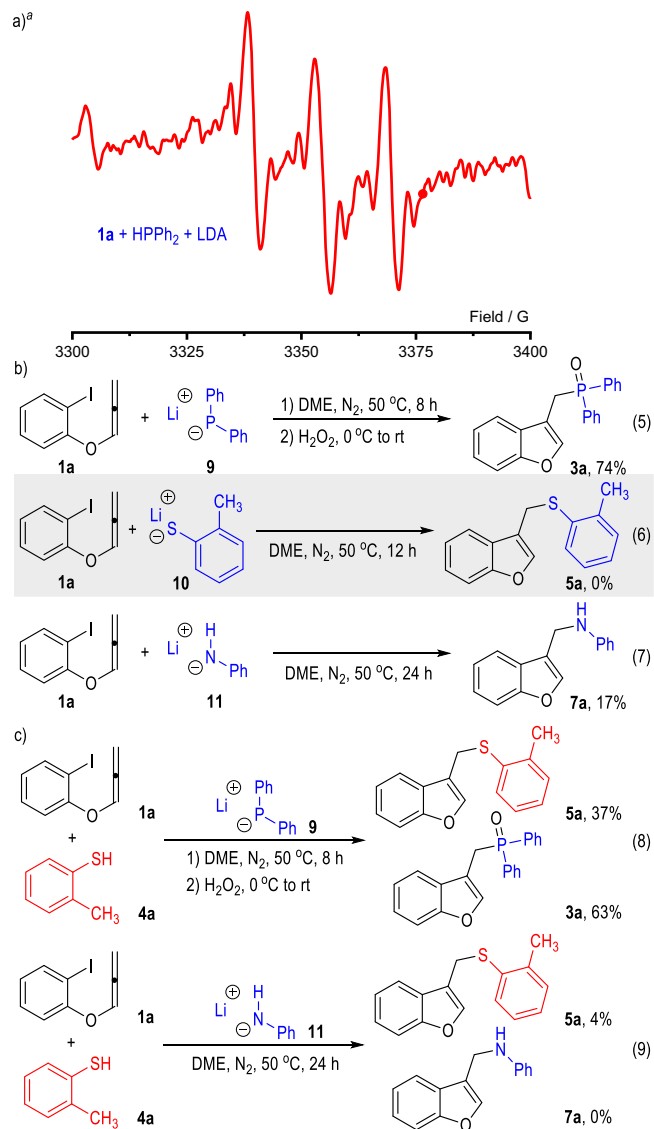

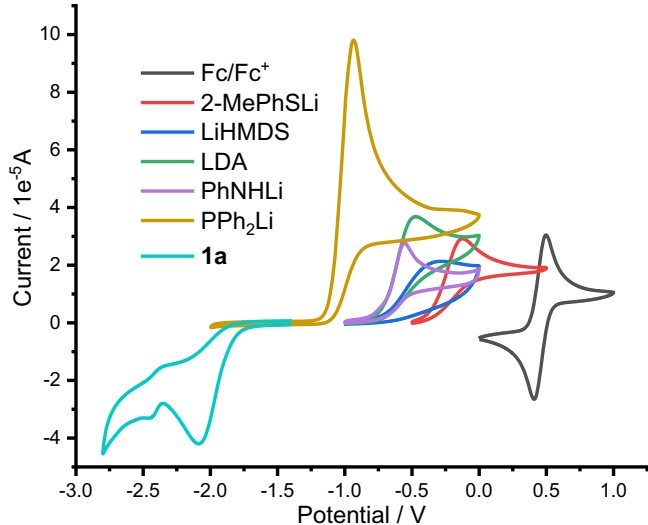

**Fig. 5 | Cyclic voltammograms of the reaction substrates.** Black line: CV of Fc/Fc⁺. Red line: CV of 2-MePhSLi. Blue line: CV of LiHMDS. Green line: CV of LDA. Purple line: CV of PhNHLi. Yellow line: CV of PPh₂Li. Cyan line CV of **1a**.

**Fig. 4 | EPR spectrum of the reaction mixtures and control experiments. a** EPR spectrum of the reaction mixtures. **b** Feasibilities of the heteroatomic anions as SEDs for the radical reactions. **c** Cross-radical coupling reactions with mercaptans. *d*The X-band EPR spectrum of 1:2:2 stoichiometric reaction of **1a** (0.1 mmol), HPPh₂ (0.2 mmol), and LDA (0.2 mmol) was measured at 298 K with DME (2 mL) as solvent at a microwave frequency of 9.418333054 GHz (*g* = 2.0023).

**Table 2 | Comparison of the redox potentials of the substrates with classic SEDs**

| Entry | Compound | Redox potential $(E_{1/2})$ $(E_{1/2})^a$ |
|---|---|---|
| 1 | PPh₂Li (**9**) | −1.03 V (oxidation)[a] |
| 2 | PhNHLi (**11**) | −0.64 V (oxidation)[b] |
| 3 | LDA | −0.60 V (oxidation)[b] |
| 4 | LiHMDS | −0.52 V (oxidation)[b] |
| 5 | 2-MePhSLi (**10**) | −0.23 V (oxidation)[a] |
| 6 | **1a** | −1.95 V (reduction)[a] |
| 7 | (**12**) | −0.38 V (oxidation)[c] |
| 8 | (TAE) | −0.78 V (oxidation)[c] |
| 9 | (TAF) | −0.82 V (oxidation)[c] |

[a]Recorded in DMF. [b]Recorded in DMSO. [c]See ref. 31.

the free amine HNR¹R². Radical coupling process between I and RS• affords the thio ether product **5**.

## Synthetic applications of the afforded 3-functionalized benzofuran

The 3-alkylsubstituted benzofuran products obtained from this radical coupling reaction are amenable through simple transformations (Fig. 7). For example, the phosphine oxide moiety of the benzofuran **3a** can be transformed into the phosphine sulfide **13** with Lawesson's reagent in almost quantitative yield[119]. The 2-position of the benzofuran ring of **3a** can be functionalized with a carbaldehyde group or a perfluoroalkyl group via simple operations, with the corresponding products **14**[120] and **15**[121] afforded moderate to good yields. A versatile Br atom can also be efficiently introduced onto the 2-position of the benzofuran ring of **3a**, with the 2-bromobenzofuran **16** given in an excellent yield[122]. The Br group in **16** is useful for various transition metal-catalyzed cross-coupling reactions. The 2-(o-tolyl)benzofuran **17**

can be afforded from **16** in 62% yield via a Suzuki reaction[123,124] and the 2-vinylbenzofuran **18** can be obtained from **16** in 63% yield via a Heck reaction[125].

## Anti-bacterial activities of the 3-functionalized benzofuran products against plant pathogens

Both the benzofuran[2] and the heteroatom[126–128] fragments of the 3-substituted benzofuran products obtained from this radical reaction have exhibited proven anti-bacterial activities against various pathogens in living organisms. We have long been devoted to the search and development of novel small molecular structures for pesticide

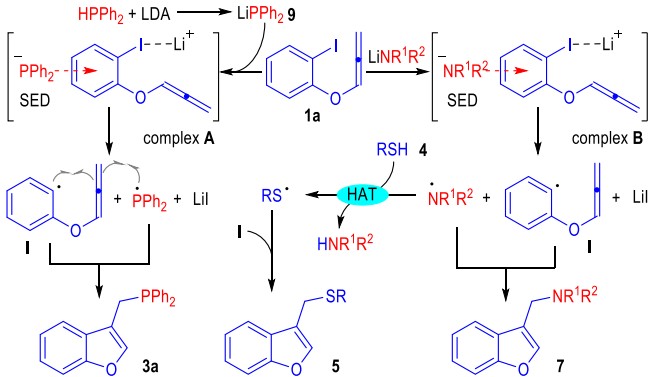

**Fig. 6 | Proposed reaction mechanism.** SED super-electron-donors, LDA lithium diisopropylamide, HAT hydrogen atom transfer.

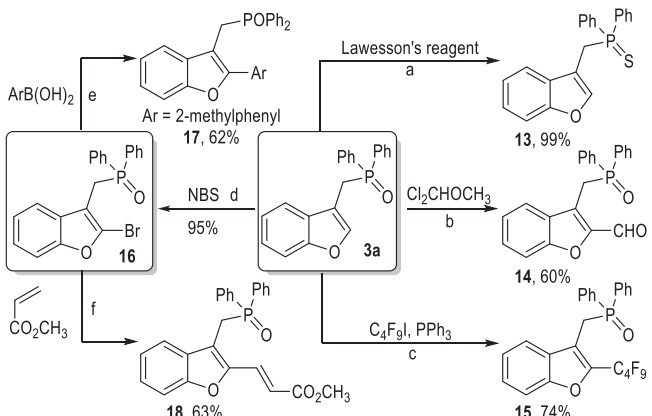

**Fig. 7 | Synthetic transformations from the benzofuran product 3a.** Conditions: **a** lawesson's reagent, toluene, 100 °C, $N_2$. **b** $Cl_2CHOCH_3$, $TiCl_4$, DCM, −10 °C to r.t., $N_2$. **c** $C_4F_9I$, $PPh_3$, Blue LED (440–445 nm, 10 W), MeOH, r.t., $N_2$. **d** NBS, $CHCl_3$:$CH_3CN$ = 1:1, −30 °C. **e** o-tolylboronic acid, $Pd(PPh_3)_4$, $K_2CO_3$, toluene:$H_2O$ (7:3), 100 °C, $N_2$. **f** Methyl acrylate, $PdCl_2$, $K_2CO_3$, NMP, 80 °C, $N_2$.

innovations and applications[129–137]. Therefore, we are very interested in their potential applications as leading structures for the development of novel pesticides. Two of the most significant plant pathogens of *Xanthomonas oryzae* pv. *oryzae* (*Xoo*)[138,139] and *Xanthomonas axonopodis* pv. *citri* (*Xac*)[140–142] were selected as the targets for the anti-bacterial activity evaluations of the afforded 3-functionalized benzofuran products (Supplementary Tables 7–10).

*Xoo* can cause leaf bite in plants and accounts for huge economic loss in rice and other crops all over the world. *Xac* is also a widespread bacterium that can cause citrus canker and severely destroy lemons, oranges, and grapefruits worldwide. The inhibitory effects of the 3-alkylbenzofuran compounds against *Xoo* and *Xac* are summarized in the Supplementary Information (Supplementary Tables 7–9). The $EC_{50}$ values of the promising anti-bacterial compounds were calculated and shown in Table 3. We noticed that the compounds **5m** and **5s** exhibited the best comprehensive anti-bacterial activities among all the compounds we tested, with the $EC_{50}$ values much lower than the commercial pesticides of BT and TC.

Based on the data from Table 3, Supplementary Tables 7–9, we can get preliminary information on the structure–activity relationship of the afforded 3-alkylbenzofuran compounds. Generally speaking, the 3-substituted benzofurans containing an S atom process better anti-bacterial activity than the ones containing an N or a P atom. Moreover, a heterocyclic substituent on the thioether moiety can help enhance the activity of the compound **5** (e.g., **5m**, **5o** vs. **5f**, **5j**, **5k**, **5s**, **5v**, **5w**). Thioethers bearing short alkyl chains (**5s**, **5v**, **5w**) showed better

**Table 3 | $EC_{50}$ values of compounds with good antibacterial activities against *Xoo* and *Xac*[a]**

| Compounds | *Xoo* | *Xac* |
|---|---|---|
|  | $EC_{50}$ (µg/mL) | $EC_{50}$ (µg/mL) |
| 5f | 17.23 ± 0.24ef | 15.67 ± 0.28e |
| 5j | 17.90 ± 3.15e | 19.25 ± 0.46d |
| 5k | 25.80 ± 1.52ab | 34.06 ± 0.88c |
| 5m | 5.88 ± 0.19j | 10.33 ± 0.11f |
| 5o | 11.07 ± 0.37i | 10.27 ± 0.21f |
| 5s | 7.38 ± 0.33j | 7.86 ± 0.16g |
| 5v | 14.01 ± 0.60gh | 13.72 ± 0.14e |
| 5w | 11.48 ± 0.90i | 11.23 ± 0.63f |
| 7d | 10.32 ± 0.40i | 9.76 ± 0.92fg |
| 7e | 21.02 ± 0.46d | 14.51 ± 0.44e |
| 7h | 15.43 ± 0.27fg | 32.03 ± 0.88c |
| 7i | 26.50 ± 1.16a | 19.02 ± 0.73d |
| 7j | 13.32 ± 0.26 h | 32.03 ± 0.88c |
| BT[b] | 23.98 ± 0.96bc | 52.67 ± 0.34b |
| TC[c] | 22.71 ± 1.26 cd | 65.10 ± 1.98a |

[a]All data were average data of three replicates. [b]BT bismerthiazol, [c]TC thiodiazole copper.

activities than the ones bearing long-chained alkyl groups (**5x**, **5y**) or aryl groups (**5f**, **5j**, **5k**). The introduction of methyl onto the 3-position of the benzene ring helps to improve the anti-bacterial activities (**5f** is better than **5a**, **5k**). Unsaturated alkyl substituents are more active than saturated alkyl substituents (**5s** vs. **5v** and **5w**).

In summary, we have developed a transition metal-free intermolecular radical coupling reaction for efficient access to 3-substituted benzofuran molecules. 2-Iodophenyl allenyl ethers and heteroatomic compounds bearing H-heteroatom (H−X) bonds are used as the reaction substrates. Strong bases such as LDA and LiHDMS are used as additives. The heteroatom anions react as SEDs with 2-iodophenyl allenyl ether substrates through SET processes. The 2-iodophenyl allenyl ether substrate can tolerate various substituents on both the benzo and the allenyl moieties. Phosphines, thiols, and anilines bearing different substitution patterns work smoothly in this transformation under similar reaction conditions. Both experimental and computational methods are used to elucidate the reaction mechanism. The afforded 3-functionalized benzofuran products showed promising practical applications in both synthetic chemistry and pesticide development. Further investigations into heteroatom anionic SEDs and the bioactivities of benzofuran derivatives are in progress in our laboratories and will be reported in due course.

## Methods
### General procedure for the preparation of 3
To a 10 mL anaerobic tube in the glove box was added **1** (0.10 mmol), $HPPh_2$ (0.20 mmol), LDA or LiHMDS (0.20 mmol), and DME (2 mL). The reaction was stirred for 8 h under $N_2$ at 50 °C. The reaction system was cooled to 0 °C and $H_2O_2$ (0.25 mmol) was added. Then the reaction system was warmed to room temperature and stirred for 1 h. Water (4 mL) was added to quench the reaction and the solution was extracted by ethyl acetate (2 mL). The organic layer was separated, washed with brine, dried over $Na_2SO_4$, and evaporated under reduced pressure. The residue was subjected to column chromatography on silica gel (petroleum ether: ethyl acetate = 3:1) to afford the pure product **3**.

### General procedure for the preparation of 5
To a 10 mL anaerobic tube in the glove box was added **1a** (0.15 mmol), **4** (0.10 mmol), LiHMDS (0.20 mmol), and DME (2 mL). The reaction

was stirred at 50 °C for 12 h under $N_2$. Water (4 mL) was added to quench the reaction and the solution was extracted by ethyl acetate (2 mL). The organic layer was separated, washed with brine, dried over $Na_2SO_4$, and evaporated under reduced pressure. The residue was subjected to column chromatography on silica gel (petroleum ether: ethyl acetate = 300:1) to afford the desired product **5**.

### General procedure for the preparation of 7

A 10 mL anaerobic tube in the glove box was added **1a** (0.10 mmol), **6** (0.10 mmol), LDA (0.20 mmol), and DME (2 mL). The reaction was stirred at 50 °C for 12 h under $N_2$. Water (4 mL) was added to quench the reaction and the solution was extracted by ethyl acetate (2 mL). The organic layer was separated, washed with brine, dried over $Na_2SO_4$, and evaporated under reduced pressure. The residue was subjected to column chromatography on silica gel (petroleum ether: ethyl acetate = 300:1) to afford the desired product **7**.

### Reporting summary

Further information on research design is available in the Nature Portfolio Reporting Summary linked to this article.

### Data availability

Crystallographic data for the structures reported in this Article have been deposited at the Cambridge Crystallographic Data Centre, under deposition numbers CCDC 2209970 (**3c**), and 2209975 (**14**). Copies of the data can be obtained free of charge via https://www.ccdc.cam.ac.uk/structures/. The full experimental details for the preparation of all new compounds, and their spectroscopic and chromatographic data generated in this study are provided in the Supplementary Information. All data are available from the authors upon request.

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

## Acknowledgements

We acknowledge financial support from the National Key Research and Development Program of China (2022YFD1700300, Z.J.). National Natural Science Foundation of China [(21961006, Z.J.), (32172459, Z.J.), (22371057, Z.J.), (22071036, Y.R.C), (21732002, Y.R.C.)]. Frontiers Science Center for Asymmetric Synthesis and Medicinal Molecules, Department of Education, Guizhou Province [Qianjiaohe KY number (2020)004, Y.R.C.]. The 10 Talent Plan (Shicengci) of Guizhou Province ([2016]5649, Y.R.C.). Science and Technology Department of Guizhou Province (Qiankehejichu-ZK[2021]Key033, Z.J.). Program of Introducing Talents of Discipline to Universities of China (111 Program, D20023, Y.R.C.) at Guizhou University. Singapore National Research Foundation under its NRF Investigatorship (NRF-NRFI2016-06, Y.R.C.) and Competitive Research Program (NRF-CRP22-2019-0002, Y.R.C.). Ministry of Education, Singapore, under its MOE AcRF Tier 1 Award (RG7/20, RG5/

19, Y.R.C.), MOE AcRF Tier 2 (MOE2019-T2-2-117, Y.R.C.), and MOE AcRF Tier 3 Award (MOE2018-T3-1-003, Y.R.C.).

## Author contributions

S.J. conducted most of the experiments. W.W., C.M. and J.Z. contributed to designs and some experiments. W.W. and G.H. conducted the DFT studies. Z.J. and Y.R.C. conceptualized and directed the project and drafted the manuscript with assistance from all co-authors. All authors contributed to discussions.

## Competing interests

The authors declare no competing interests.
