## [Peer Review File · Nature Communications]

REVIEWER COMMENTS

Reviewer #1 (Remarks to the Author):

Hao, Chi and coworkers present a new series of super electron donors. The strategy employed here involves deprotonation of an X-H bond to give the reducing anion. That anion undergoes SET with the aryl group of the 2-iodophenyl allenyl ether substrate. Loss of iodide generates an aryl radical that cyclizes to give a new radical that couples with the X• to give the product. This same strategy has been used (Fig 1, eq 4) with a different reducing agent. However, I would argue that the reducing agents used here, which eventually become coupling partners in the radical-radical coupling process, are quite different and broaden the scope of this approach significantly. Thus, the novelty of this work, in my opinion, is high.

Table 1 outlines the optimization where Li⁺ derived bases are better than Na and K. Any idea why? Even a hypothesis would be useful to the reader. Unfortunately, the phosphine products seem to be sensitive to air and the phosphine must be oxidized. That is surprising to me. Is the problem the P-center or benzofuran or the combination of the two? I thought diphenylmethyl phosphine was pretty air stable.

As shown in Schemes 2 and 3, thiols and amines also work as SED's. This is a nice addition to the paper. Does the reaction with thiols work completely in the absence of light? I recall that thiolates for complexes that can undergo light activated SET reactions with different substrates. Do the authors observe any disulfide, derived from radical-radical coupling? The amine products still possess an acidic NH. Do they undergo a second reaction with the substrate?

In the mechanistic work, BHT will just get deprotonated rapidly. I do not think the results with BHT mean anything. TEMPO is an oxidant and LDA can act as a reductant. Thus, I worry that reaction of LDA with TEMPO will consume the LDA needed to promote the reaction, causing the yield to be reduced (as observed).

Do any of these reactions work with aryl bromides or chlorides?

SI: some of the starting materials are known. Do the obtained spectra match the published data? Give yields in weights of compound and mmol products. What solvent system was used in each purification?

Most of the compounds look clean by NMR, but there are a lot that have tall peaks around 2 (water?). Some compounds, like 3c, 5b 17, need to be purified again. More detail could be added to the SI. See some published works as models.

,

Minor points.

There were several corresponding authors on the paper that is listed as by Walsh. Please add all the corresponding authors' names, like the scheme.

Spelling issue: Notably, the 2-azaallynyl anion. Should be 2-azaallyl I believe.

2-iodophenol allenyl ethers should be 2-iodophenyl allenyl ethers

The strengths of this manuscript are the use of a series of deprotonated compounds that behave as super electron donors. This work has been presented in the context of the synthesis of benzofurans, which are important heterocycles. The weakness is that the synthetic strategy is similar to prior work (same substrates) but I do not view this as a significant issue. There are several questions above that need to be answered in the text, but if the authors can successfully address these questions, I would recommend publication in NComm. This is a nice paper that will attract the attention of the medicinal chemistry and synthetic communities

Reviewer #2 (Remarks to the Author):

Recommendation: Publication after revision rather than current manuscript.

Comments:

In this work, Jiang and co-workers reported a systematic radical reactions for facile synthesis of 3-substituted benzofurans. Strong Lewis bases such as LDA and LiHDMS are used as the base to deprotonate the diphenylphosphine, thiols and analines. Both experimental and computational methods are used to elucidate the reaction mechanism. In the detail of reaction mechanism, there exit some indistinct results which should be further explored.

1. In the supporting information, the proposed mechanism is including two possible pathways: triplet state pathway and singlet state pathway. However, the detailed calculation of singlet state pathway not provide. The comparison of triplet state and singlet state pathways should display.
2. In the mechanism, more comprehensive possible mechanism should be considered rather than one.
3. There are steps: (1) cations exchange between HPPh2 and LDA; (2) electron transfer from phosphor to benzene ring; and (3) intersystem crossing and quenchin. However, the information between in text (Figure 5) and supporting information are not consistent.
4. In Figure 5 and Figure S1, some geometries are not consistent, such as TS2. In addition, unified notation should be used in in text (Figure 5) and supporting information (Figure S1), such as TS2T used in text and SI.
5. In the supporting information, in line 161 of paragraph S13, "triplet state pathway" change to "singlet state pathway".

Reviewer #3 (Remarks to the Author):

Comments:

The entitled manuscript "Facile Access to Benzofuran Derivatives through Radical Reactions with Heteroatom-centered Super-electron-donors" disclosed the reaction of aryl allenyl ethers with phosphines, thiols and anilines to produce benzofurans derivatives. This reaction gave benzofurans derivatives with generally good yields and substrate tolerance, however, there are still some features that slightly reduce the importance of this contribution.

First of all, its novelty is not high enough. The preparation of benzofurans derivatives through single electron transfer (SET) process between aryl allenyl ethers and super electron donors (SEDs) has been reported. Although this work provides benzofuran molecules bearing 3-phospha- / thio- / aza-methyl groups, this method does not constitute the significant advancement in terms of concept and mechanism of the reaction.

Moreover, the evidences of heteroatom anions as a super electron donor were insufficient, and the authors need to do more experiments to verify the mechanism. The mechanistic studies presented in the submitted article are too underpowered to convince the reader.

In addition, the manuscript will be better if more discussion were added, but not simply replicate the content of tables and schemes.

Overall, the mechanistic studies presented in the submitted article are too underpowered to merit publication as is in Nature Communications. If the authors address the concerns listed below, however, it is very conceivable that the resulting manuscript would rise up to the acceptable standard.

Some suggestions and questions for improvement of the manuscript are given here below:

1. In order to better verify that heteroatom anions possess the characteristics of super electron donor, cyclic voltammetry should be tested to indicate the redox state regimes of the phosphines, thiols and anilines.
2. In the mechanism study, the authors need to do experiments to verify the existence of nitrogen / phosphorus / sulfur radical intermediates in the reaction process, such as radical clock experiment and electron paramagnetic resonance (EPR) experiment.
3. Phosphines, thiols and anilines were applied to this systems, does alcohol do the same to give benzofurans derivatives? It is not essential that these studies are included in the manuscript but if they don't work then that would be of interest to the reader.
4. The solvent screening studies are also incomplete. Several studies indicate that DMSO and DMF are superior solvents for electron-transfer processes. Results from these two solvents should be included in Table 1.
5. In the scheme 3, substitutions on the 2-position of the aniline bearing electron-withdrawing led to significant drops on the product yields (7f). What happens if substitutions on the 2-position of the aniline bearing electron-donating?
6. In the figure 3, "BH₃?THF" should be "BH₃•THF"; the yield of compound 5a is wrong.
7. In the main text, compounds 7g, 3c, and 13 should be bolded on pages 4 and 7.
8. In the supporting information, "triplet" in line 161 should be "singlet".

Reviewer #4 (Remarks to the Author):

The manuscript reports the antibacterial activity of selected benzofurans against *Xanthomonas oryzae* pv. *oryzae* and *Xanthomonas axonopodis* pv. *citri*. The authors use thiodiazole-copper and bismertiazol, two known agricultural antibacterial agents, as positive controls and demonstrate that their compounds have higher inhibition rates at 50 and 25 µg/mL than these two known compounds. However, it is not clear why the incubation time was in the range of 38 to 48 hours. All compounds and all three replicates should be tested after (preferably) 48 h of incubation time.

Moreover, given the rather high standard deviations reported in Table 3 and Table 4, the antibacterial activity at 50 µg/mL and 25 µg/mL is comparable. Therefore, the authors should repeat the experiments and present the dose-response data (determination of the EC₅₀ value) or alternatively, determine the MIC values, i.e. the lowest concentration that leads to an inhibition rate of more than 90%.

The authors claim that their compounds have a higher bactericidal effect than the commercial drugs. This is an over-interpretation of the results for the reasons mentioned above. Furthermore, the test used by the authors does not allow to determine whether the activity is bacteriostatic or bactericidal. This should be clarified in the text or substantiated by additional tests.

After determining the EC₅₀ or MIC values, the authors should briefly discuss the structure-activity relationship, highlighting the structural features that are critical for strong antibacterial activity.

In addition, all compounds tested for biological activity should be greater than 95% pure. The authors should provide the purity data of the tested compounds.

The authors only performed the phenotypic screening. Do they have any clues or evidence for the bacterial target whose inhibition leads to the observed effects? Do they have any data on the cytotoxicity of the compounds against human cells? This could give an indication of whether the effect of the compounds is non-specific.

Reviewer #1 has recommended acceptance of our manuscript after minor revisions.

- a) **Reviewer's comment:** "Table 1 outlines the optimization where Li⁺ derived bases are better than Na and K. Any idea why? Even a hypothesis would be useful to the reader."

Our Response: The positively charged Li⁺ is proposed to react as a Lewis acid in the reaction system to activate the aryl iodide substrate **1**. The acidity of Li⁺ is stronger than the acidities of Na⁺ and K⁺. Therefore, the Li⁺ derived bases can promote the radical reaction better than Na⁺ and K⁺. This hypothesis has been added in the revised manuscript in Paragraph 1 on Page 3 as "This is probably due to the increased acidity of the positively charged Li⁺ existed in the reaction system for a better activation of the phenyl iodide substrate **1a**."

- b) **Reviewer's comment:** "Unfortunately, the phosphine products seem to be sensitive to air and the phosphine must be oxidized. That is surprising to me. Is the problem the P-center or benzofuran or the combination of the two? I thought diphenylmethyl phosphine was pretty air stable."

Our Response: We have carried out control experiments to systematically compare the stabilities of the benzofurymethyldiphenylphosphine product **2a** and the diphenylmethyl phosphine. Stark contrast could be observed in their deterioration rates in THF-d⁸ after a certain period (Figure R1).

The less stability of the product **2** than the diphenylmethyl phosphine is believed to be attributed to the strong electron-donating benzofuran group existed in the product structure. The electron density on the P atom of the product **2** could be increased by the benzofuran substituted methylene group, so that the reducing ability of the product **2** is stronger than the diphenylmethyl phosphine and could be more easily oxidized in air conditions.

This information has been added in the revised Supporting Information on Pages S6 to S7.

Figure R1. ³¹P NMR analysis of the product **2a** and the diphenylmethyl phosphine.

- c) **Reviewer's comment:** "Does the reaction with thiols work completely in the absence of light?"

Our Response: We have carried out control experiments with thiols in the absence of light (Figure R2). The reaction worked smoothly without any light irradiation. The product yield did not change under dark condition. This information has been added in the revised manuscript in Scheme 2, footnote b.

Figure R2. Control experiments in dark condition.

- d) **Reviewer's comment:** "Do the authors observe any disulfide, derived from radical-radical coupling?"

Our Response: We do have observed the formation of the disulfide intermediates at the beginning of the radical coupling reaction. Moreover, we have carried out control experiments to subject the thiophenol substrate **4a** to the basic reaction system (Figure R3). The disulfide **4a'** was isolated in a moderate yield and characterized with ^1H NMR, ^{13}C NMR and GC-MS. This information has been added in the revised manuscript in Paragraph 2 on Page 4, and in the revised Supporting Information on Pages S11 to S13.

Figure R3. Characterization of the disulfide by-product.

- e) Reviewer's comment: "The amine products still possess an acidic NH. Do they undergo a second reaction with the substrate?"

Our Response: The primary amine substrate **6a** can go through a two-stepped radical coupling reaction to give the tertiary amine product **7a'** in 6% yield (Figure R4). The secondary amine product **7a** can also react again under the reaction condition and give the **7a'** in 20% yield. This information has been added in the revised manuscript in Scheme 3 and Paragraph 4 on Page 4, and in the revised Supporting Information on Page S78.

Figure R4. Generation of the tertiary amine product **7a'** through a one-pot or a 2-stepped operation.

- f) **Reviewer's comment:** "In the mechanistic work, BHT will just get deprotonated rapidly. I do not think the results with BHT mean anything. TEMPO is an oxidant and LDA can act as a reductant. Thus, I worry that reaction of LDA with TEMPO will consume the LDA needed to promote the reaction, causing the yield to be reduced (as observed)."

Our Response: We have deleted the control experiments using radical scavengers in the revised manuscript. Instead, we have added the mechanistic investigation experiment with EPR spectral analysis, cross-reactions and the CV results. This information has been added on Pages 5 to 7 in the revised manuscript, as highlighted in yellow.

- g) **Reviewer's comment:** "Do any of these reactions work with aryl bromides or chlorides?"

Our Response: None of these reactions work with aryl bromides or chlorides, which is probably due to the less oxidation abilities of aryl bromides or chlorides. This information has been added in the revised manuscript in Paragraph 1 on Page 4.

- h) **Reviewer's comment:** "SI: some of the starting materials are known. Do the obtained spectra match the published data? Give yields in weights of compound and mmol products. What solvent system was used in each purification?"

Our Response: We have checked the ^1H , ^{13}C NMR spectra and HRMS of all the starting materials and compared them with the literature reports. All the spectra we obtained match with the published data. This information has been added in the revised Supporting Information on Pages S52 to S57 and Pages S84 to S96.

The yields of all the compounds have been given in weights and mmols in the revised Supporting Information on Pages S52 to S57.

All the solvent we used for the column purification have been given in the revised Supporting Information on Pages S52 to S57.

- i) **Reviewer's comment:** "Most of the compounds look clean by NMR, but there are a lot that have tall peaks around 2 (water?). Some compounds, like **3c**, **5b** **17**, need to be purified again. More detail could be added to the SI. See some published works as models."

Our Response: The tall peak around 2 is from water. Compounds **3c**, **5b** and **17** have been purified again. We have added the detailed experimental procedures in the revised Supporting Information on Pages S57 to S82 and Pages S99 to S179.

- j) **Reviewer's comment:** "There were several corresponding authors on the paper that is listed as by Walsh. Please add all the corresponding authors' names, like the scheme."

Our Response: We have made this change in the revised manuscript in Paragraph 2 on Page 1.

- k) **Reviewer's comment:** "Spelling issue: Notably, the 2-azaallyl anion. Should be 2-azaallyl I believe."

Our Response: We have corrected this error in the revised manuscript in Paragraph 2 on Page 1.

l) Reviewer's comment: "2-iodophenol allenyl ethers should be 2-iodophenyl allenyl ethers."

Our Response: We have corrected this error throughout the revised manuscript.

Reviewer #2 has recommend publication of this paper after certain revisions on the DFT studies.

a) Reviewer's comment: "In the supporting information, the proposed mechanism is including two possible pathways: triplet state pathway and singlet state pathway. However, the detailed calculation of singlet state pathway not provide. The comparison of triplet state and singlet state pathways should display."

Our Response: We have added the detailed calculation of singlet state pathway and compared it with the triplet state pathway in the revised Supporting Information on Pages S13 to S14.

b) Reviewer's comment: "In the mechanism, more comprehensive possible mechanism should be considered rather than one."

Our Response: We have added the alternative singlet state pathway in the revised manuscript in Figure 4 and in Paragraph 5 on Page 5.

c) Reviewer's comment: "There are steps: (1) cations exchange between HPPPh₂ and LDA; (2) electron transfer from phosphor to benzene ring; and (3) intersystem crossing and quenchin. However, the information between in text (Figure 5) and supporting information are not consistent."

Our Response: We have re-examined the reaction mechanism provided in the Supporting Information and made necessary verifications to ensure it is consistent with that stated in the manuscript.

d) Reviewer's comment: "In Figure 5 and Figure S1, some geometries are not consistent, such as TS2. In addition, unified notation should be used in in text (Figure 5) and supporting information (Figure S1), such as TS2T used in text and SI."

Our Response: We have revised the corresponding geometries of the figures and the notations of the structures in the Supporting Information to make it consistent with that stated in the manuscript.

e) Reviewer's comment: "In the supporting information, in line 161 of paragraph S13, "triplet state pathway" change to "singlet state pathway"."

Our Response: We have corrected this error in the revised Supporting Information.

Reviewer #3 has recommend publication of this paper after further clarification on the reaction mechanism. We have carried out plenty of control experiments, spectral analysis, and cyclic voltammetry examination to shed light on the detailed reaction mechanism.

a) Reviewer's comment: "In order to better verify that heteroatom anions possess the characteristics of super electron donor, cyclic voltammetry should be tested to indicate the redox state regimes of the phosphines, thiols and anilines."

Our Response: We have tested the redox state regimes of the lithium salts derived from the phosphines, thiols and anilines used in our reactions. The results were shown in the revised manuscript in Table 2 on Page 6 and Paragraph 3 on Page 7.

The oxidation potentials of the lithium diphenylphosphanide and the lithium amides are comparable to the oxidation potentials of the classic SED molecules that are efficient in the reduction of aryl iodides via SET processes. Although the oxidation potentials of the lithium salts were less negative than the reduction potential of the substrate **1a**, the irreversibility of the elimination of the iodide anion from **1a** contributed to the driving force for the single electron transfer reaction from the P- and N-centered anions to the substrate **1a**.

Therefore, we believe that the anions generated from the phosphines and anilines can react directly as the SEDs, while the mercaptan substrate **4** can go through a hydrogen-atom-transfer (HAT) reaction with the amino radical $\cdot\text{NR}^1\text{R}^2$ to give the more stable thio radical $\text{RS}\cdot$ and finally give the thio ether product **5**.

- b) Reviewer's comment: "In the mechanism study, the authors need to do experiments to verify the existence of nitrogen / phosphorus / sulfur radical intermediates in the reaction process, such as radical clock experiment and electron paramagnetic resonance (EPR) experiment."

Our Response: We have carried out the EPR experiment to verify the generation of radical species in the reaction system (Figure R5). The EPR spectrum of the mixture of **1a**, HPPPh₂ and LDA in DME at 25 °C exhibited a signal at approximately $g = 2.0023$, which is similar to the signal of the g factor of a phenyl radical. The EPR experiments have clearly demonstrated the generation of radical intermediates in our reactions.

Figure R5. EPR spectra of the reaction mixtures.

In addition, we have been devoted into the synthesis of a diversity of reaction substrates for radical clock experiments for over 2 months. Unfortunately, none of these approaches provided the desired compound (Figure R6). For instance, the isomerization of the alkyne molecule **s1** gave the phenol **s2** or the de-halogenated product **s3** in moderate to good yields without formation of the target allene product **s4**. The base-promoted isomerization of the alkyne **s5** provided the conjugated triene product **s6** in a good yield, with no formation of the target allene product **s7** observed. This information has been added in the revised Supporting Information on Page S33.

Figure R6. Failed experiments on the substrate synthesis for radical clock experiment.

- c) Reviewer's comment: "Phosphines, thiols and anilines were applied to this systems, does alcohol do the same to give benzofurans derivatives? It is not essential that these studies are included in the manuscript but if they don't work then that would be of interest to the reader."

Our Response: We have examined a diversity of alcohols and phenols as the potential radical precursor for this transformation. However, none of the alcohols or phenol substrates gave the desired benzofuran products. We assume that the oxygen radicals are not as stable as the phosphine-, sulfur- or nitrogen-centered radicals. Therefore, the deprotonated oxide anions cannot give radicals that are stable enough to react with the aryl radical intermediates in the reaction system. This information has been added in the revised manuscript in Paragraph 2 on Page 5.

- d) Reviewer's comment: "The solvent screening studies are also incomplete. Several studies indicate that DMSO and DMF are superior solvents for electron-transfer processes. Results from these two solvents should be included in Table 1."

Our Response: We have systematically re-examined the reaction solvents. The results of the reaction carried out in DMF and DMSO have been included in Table 1, Entries 11 and 12 in the revised manuscript. The results of the detailed solvent screening have been added in the revised supporting information on Page S5.

- e) Reviewer's comment: "In the scheme 3, substitutions on the 2-position of the aniline bearing electron-withdrawing led to significant drops on the product yields (7f). What happens if substitutions on the 2-position of the aniline bearing electron-donating?"

Our Response: We have used the aniline bearing an electron-donating 2-methyl group as the substrate to test the reaction. The target product **7g** was given in 20% isolated yield. We assume that the drops in the product yield might be resulted from the steric hindrance on the nitrogen radicals caused by the 2-substituents. This information has been added in the revised manuscript in Scheme 3 and Paragraph 3 on Page 4.

- f) Reviewer's comment: "In the figure 3, "BH3?THF" should be "BH3•THF"; the yield of compound 5a is wrong."

Our Response: We have changed "BH3?THF" into "BH3•THF". The yield of the compound **5a** is calculated based on the amount of the mercaptan substrate **4**. We used **1a** 38.71 mg (0.15 mmol), **4a** 12.42 mg (0.10 mmol) with 0.20 mmol of LiHMDS and 2 mL DME at 50 °C for 12 h under N₂. The product **5a** is obtained in 24.16 mg (0.095 mmol). The yield of **5a** is 95% (0.095 mmol / 0.1 mmol).

- g) Reviewer's comment: "In the main text, compounds 7g, 3c, and 13 should be bolded on pages 4 and 7."

Our Response: We have made the compounds **7g**, **3c**, and **13** bold on Pages 4 and 7.

- h) Reviewer's comment: "In the supporting information, "triplet" in line 161 should be "singlet"."

Our Response: We have made this revision in the revised supporting information on Page S13.

Reviewer #4

- a) Reviewer's comment: "However, it is not clear why the incubation time was in the range of 38 to 48 hours. All compounds and all three replicates should be tested after (preferably) 48 h of incubation time."

Our Response: The best time for the anti-bacterial activity test falls in the logarithmic growth stage of cells. In the logarithmic growth stage, the cells grow in a rapid rate since they have already experienced a short adaptation period and gained plenty of nutrition. The turbidity value of the bacterial solution in the control experiment ranges from 0.6 to 0.8. For *Xac*, it takes about 38 h to reach the logarithmic growth stage, and the logarithmic growth stage will end after 48 h. For *Xoo*, it takes about 48 h to reach the logarithmic growth stage.

Based on the reasons stated above, we did the bioactivity test in the range of 38 to 48 h.

- b) Reviewer's comment: "Moreover, given the rather high standard deviations reported in Table 3 and Table 4, the antibacterial activity at 50 µg/mL and 25 µg/mL is comparable. Therefore, the authors should repeat the experiments and present the dose-response data (determination of the EC50 value) or alternatively, determine the MIC values, i.e. the lowest concentration that leads to an inhibition rate of more than 90%."

Our Response: We have re-examined the bioactivities of all the products obtained from our method. The EC₅₀ data of the highly active compounds has been calculated and provided in Table 3 of the revised manuscript. All the updated data of the primary screening has been provided in the revised Supporting Information on Pages S45 to S50.

- c) Reviewer's comment: *"The authors claim that their compounds have a higher bactericidal effect than the commercial drugs. This is an over-interpretation of the results for the reasons mentioned above."*

Our Response: We have rephrased our expressions in Paragraph 2 on Page 8 of the revised manuscript as "We noticed that the compounds **5m** and **5s** exhibited the best comprehensive anti-bacterial activities among all the compounds we tested, with the EC₅₀ values much lower than the commercial pesticides of BT and TC."

- d) Reviewer's comment: *"Furthermore, the test used by the authors does not allow to determine whether the activity is bacteriostatic or bactericidal. This should be clarified in the text or substantiated by additional tests."*

Our Response: We thank this reviewer for his valuable suggestion.

The bioactivity test in this manuscript aims to broaden the application scope of the synthetic method. We demonstrated that several of the 3-substituted benzofuran products obtained from our radical reactions possessed promising anti-bacterial activities against plant pathogens. Some of them showed even lower EC₅₀ values than the commercial drugs of TC and BT.

We believe that both bacteriostatic and bactericidal effects contributed to the anti-bacterial bioactivities of our products, although it is extremely difficult for us to clarify the exact mechanism.

- e) Reviewer's comment: *"After determining the EC₅₀ or MIC values, the authors should briefly discuss the structure-activity relationship, highlighting the structural features that are critical for strong antibacterial activity."*

Our Response: The preliminary analysis on the structure-activity-relationship has been added in the revised manuscript on Page 8, Paragraph 3, stated as "Based on the data from Tables 3, S7, S8, and S9, we can get preliminary information on the structure-activity-relationship of the afforded 3-alkylbenzofuran compounds. Generally speaking, the 3-substituted benzofurans containing an S atom process better anti-bacterial activity than the ones containing an N or a P atom. Moreover, a heterocyclic substituent on the thioether moiety can help enhance the activity of the compound **5** (e.g., **5m**, **5o** v.s. **5f**, **5j**, **5k**, **5s**, **5v**, **5w**). Thioethers bearing short alkyl chains (**5s**, **5v**, **5w**) showed better activities than the ones bearing long-chained alkyl groups (**5x**, **5y**) or aryl groups (**5f**, **5j**, **5k**). The introduction of a methyl at the 3-position of the benzene ring helps to improve the anti-bacterial activities (**5f** is better than **5a**, **5k**). Unsaturated alkyl substituents are more active than saturated alkyl substituents (**5s** v.s. **5v** and **5w**)."

- f) Reviewer's comment: *"In addition, all compounds tested for biological activity should be greater than 95% pure. The authors should provide the purity data of the tested compounds."*

Our Response: We have examined the purities of all the compounds we used in the bioactivity test through ¹H NMR analysis. All the compounds possessed purities greater than 95%. We have added this information in the revised Supporting Information on Page S45, Paragraph 2.

- g) Reviewer's comment: *"The authors only performed the phenotypic screening. Do they have any clues or evidence for the bacterial target whose inhibition leads to the observed effects? Do they have any data on the cytotoxicity of the compounds against human cells? This could give an indication of whether the effect of the compounds is non-specific."*

Our Response: We have done the biological test to verify the potential application of our products in the field of pesticide development. At present, we were not able to clarify the target proteins of our compound in the bacterium. The investigation into the pesticide target will be carried out after a pesticide candidate is figured out, whose EC₅₀ value should be lower than 1 µg/mL.

Moreover, to verify whether the compounds **5m**, **5s** and **7d** had potential mammalian toxicity, the structures of benzofuran and the compounds **5m**, **5s** and **7d** were submitted to the website (<https://admetmesh.scbdd.com/>). The predicted results showed that the toxicities of the 3-substituted benzofurans were significantly lower than the benzofuran. The results are displayed as follows:

(1) The hERG Blockers, Human Hepatotoxicity and Maximum Recommended Daily Dose (FDAMDD+) of all the benzofuran-derived molecules are low (**5m**: 0.208, 0.149, 0.083; **5s**: 0.014, 0.078, 0.046; **7d**: 0.081, 0.446, 0.261; benzofuran: 0.083, 0.058, 0.083).

(2) The AMES Toxicity and Eye Corrosion of **5m**, **5s** and **7d** are lower than the benzofuran (**5m**: 0.123, 0.016; **5s**: 0.106, 0.138; **7d**: 0.021, 0.007; benzofuran: 0.311, 0.912).

(3) The Rat Oral Acute Toxicity of the compound **5m**, **5s** and **7d** is lower than the benzofuran (**5m**: 0.689; **5s**: 0.112; **7d**: 0.482; benzofuran: 0.891).

This part has been added in the revised Supporting Information on Pages S51.

REVIEWERS' COMMENTS

Reviewer #1 (Remarks to the Author):

Jin, Hao, Chi and their co-workers have revised their manuscript on SED. This is the second review, and the authors have answered most of my questions to my satisfaction. They have put in substantial work to revise the manuscript. I cannot evaluate the bioactivity studies and believe that the manuscript is suitable for publication even without them.

The revised mechanism makes more sense to me and it a nice addition.

Minor points:

considerable attentions.27-39 change to considerable attention.27-39.

With an optimal reaction condition change to With optimal reaction conditions

To probe insights into the reaction mechanism, change to To probe the reaction mechanism,

Alternatively, an iron-pair transfer change to Alternatively, an ion-pair transfer

formation of the iron pair change to formation of the ion pair

is less possible change to is less likely

clock experiments were failed changed to clock experiments failed

phenyliodide substratre. Spelling.

transmetal-catalyzed change to transition metal-catalyzed. Fix in several places.

Strong Lewis bases such as LDA But LDA is not a Lewis acid. Li⁺ is a Lewis acid. LDA is a Bronsted base. So Strong Lewis bases such as LDA and LiHDMS are used as the basic additives. Should become: Strong bases such as LDA and LiHDMS are used as additives.

on both the phenyl and the allenyl moieties change to on both the benzo and the allenyl moieties.

Analines should be anilines.

I suggest adding the following reference: (1) Gentner, T. X.; Mulvey, R. E. Alkali-Metal Mediation: Diversity of Applications in Main-Group Organometallic Chemistry. *Angew. Chem. Int. Ed.* 2021, 60 (17), 9247-9262. DOI: <https://doi.org/10.1002/anie.202010963>. This review documents how dramatically different the reactivity of species with different main group metals can be.

In sum, this is fascinating chemistry. There are a series of very minor changes that need to be made before publication in Nature Comm. The paper does not need to be reviewed again but should be published without delay.

Reviewer #2 (Remarks to the Author):

I am satisfied with the improved version of this manuscript and I recommend publishing it in its present shape.

Reviewer #3 (Remarks to the Author):

Comments:

The authors have addressed all the questions commented by reviewers. Detailed mechanistic experiments have been added in the revised manuscript. The manuscript is recommended for publication in Nature Communications as its current version. Some detail revisions are suggested:

- (1) In Paragraph 1 on Page 3, "entry 11" should be "entry 13" and "entry 12" should be "entry 14".
- (2) In Figure 3, the format of "BH₃•THF" is still wrong.
- (3) In "Mechanistic study" on Page 5, the number of "phosphine anion intermediate I" is wrong.
- (4) In eq. 5 in Figure 6, "3a" should be "7a".

Reviewer #4 (Remarks to the Author):

The authors have satisfactorily addressed my comments. I support the publication of the manuscript.

Reviewer #1 has recommended acceptance of our manuscript after minor revisions.

a) Reviewer's comment: "*considerable attentions.27-39 change to considerable attention.27-39.*"

Our Response: We have changed "considerable attentions" into "considerable attention".

b) Reviewer's comment: "*With an optimal reaction condition change to With optimal reaction conditions*"

Our Response: We have changed "With an optimal reaction condition" into "With optimal reaction conditions".

c) Reviewer's comment: "*To probe insights into the reaction mechanism, change to To probe the reaction mechanism*"

Our Response: We have changed "To probe insights into the reaction mechanism" into "To probe the reaction mechanism".

d) Reviewer's comment: "*Alternatively, an iron-pair transfer change to Alternatively, an ion-pair transfer*"

Our Response: We have changed "iron-pair" into "ion-pair".

e) Reviewer's comment: "*formation of the iron pair change to formation of the ion pair*"

Our Response: We have changed "iron-pair" into "ion-pair".

f) Reviewer's comment: "*is less possible change to is less likely*"

Our Response: We have changed "possible" into "likely".

g) Reviewer's comment: "*clock experiments were failed changed to clock experiments failed*"

Our Response: We have changed "clock experiments were failed" in to "clock experiments failed".

h) Reviewer's comment: "*phenyliodide substratre. Spelling.*"

Our Response: We have changed "substratre" into "substrate".

i) Reviewer's comment: "*transmetal-catalyzed change to transition metal-catalyzed. Fix in several places.*"

Our Response: We have changed "transmetal-catalyzed" into "transition metal-catalyzed" throughout the manuscript.

j) Reviewer's comment: "*Strong Lewis bases such as LDA But LDA is not a Lewis acid. Li+ is a Lewis acid. LDA is a Bronsted base. So Strong Lewis bases such as LDA and LiHDMS are used as the basic additives. Should become: Strong bases such as LDA and LiHDMS are used as additives.*"

Our Response: We have changed "Strong Lewis bases such as LDA and LiHDMS are used as the basic additives" into "Strong bases such as LDA and LiHDMS are used as additives".

k) Reviewer's comment: "*on both the phenyl and the allenyl moieties change to on both the benzo and the allenyl moieties.*"

Our Response: We have changed "on both the phenyl and the allenyl moieties" into "on both the benzo and the allenyl moieties".

l) Reviewer's comment: "Analines should be anilines."

Our Response: We have changed "analines" into "anilines".

m) Reviewer's comment: "I suggest adding the following reference: (1) Gentner, T. X.; Mulvey, R. E. Alkali-Metal Mediation: Diversity of Applications in Main-Group Organometallic Chemistry. *Angew. Chem. Int. Ed.* 2021, 60 (17), 9247-9262. DOI: <https://doi.org/10.1002/anie.202010963>. This review documents how dramatically different the reactivity of species with different main group metals can be."

Our Response: We have added this reference as the ref. 86 in the revised manuscript. We have also rectified the reference numbers throughout the manuscript.

Reviewer #2 has recommend publication of this paper without alterations.

Reviewer #3 has recommended acceptance of our manuscript after minor revisions.

a) Reviewer's comment: "In Paragraph 1 on Page 3, "entry 11" should be "entry 13" and "entry 12" should be "entry 14."

Our Response: We have rearranged the order of the entries 10 to 12 in Table 1 and have rectified the entry numbers in Paragraph 3 on Page 2 in the revised manuscript.

b) Reviewer's comment: "In Figure 3, the format of "BH₃•THF" is still wrong."

Our Response: We have rectified the format of "BH₃•THF" in Figure 2d in the revised manuscript.

c) Reviewer's comment: "In "Mechanistic study" on Page 5, the number of "phosphine anion intermediate I" is wrong."

Our Response: We have changed "phosphine anion intermediate I" into "phosphine anion intermediate Ph₂P⁻" in the revised manuscript on Page 5.

d) Reviewer's comment: "In eq. 5 in Figure 6, "3a" should be "7a"."

Our Response: We have changed "3a" should be "7a" in the revised manuscript in Figure 4, eq. 9.

Reviewer #4 has recommend publication of this paper without alterations.